# Non-linear associations of a body shape index with diabetes among adults: A cross-sectional study

Kaiqi Chen[1☯], Cai Tang[2☯], Yunhua Li[3], Danping Zhu[2], Xijian Zhang[2], Shikui Cui[2], Zhixi Zhu[2], Fang Fang[2]*

1 Hospital of Chengdu University of Traditional Chinese Medicine, Chengdu, China, 2 Department of Endocrinology, Chongqing City Hospital of Traditional Chinese Medicine, Chongqing, China, 3 College of Education, Chengdu College of Arts and Sciences, Chengdu, China

☯ These authors contributed equally to this work.

* fnf7703@163.com

## Abstract

### Objective

This study aimed to explore the relationship between a body shape index (ABSI) and diabetes, and to assess the robustness of this relationship across different population subgroups.

### Methods

Utilizing data from the National Health and Nutrition Examination Survey (NHANES) database, this study employed multivariate linear regression analysis to evaluate the association between ABSI and the likelihood of having diabetes. This research further explored non-linear issues in gender stratification through smooth curve fitting and two-part linear regression models, analyzing different subgroups including gender, race, hypertension, and stroke.

### Results

A total of 34,693 participants were involved in the study, with a diabetes prevalence of 11.60%. The prevalence increased with higher tertiles of ABSI. After comprehensive adjustment, ABSI was positively correlated with diabetes (OR = 1.42, 95% CI: 1.33, 1.52). Participants in the highest quartile of ABSI had a 96% higher odds of having diabetes (OR = 1.96, 95% CI: 1.69, 2.26), than did those in the lowest quartile. Smooth curve fitting analysis revealed a non-linear, inverse L-shaped relationship between ABSI and diabetes, with a breakpoint at 9.54. Subgroup analyses indicated that the association between ABSI and diabetes remained stable across different populations, except for those with a history of stroke.

**Data availability statement:** Publicly available datasets were analyzed in this study. This data can be found here: https://www.cdc.gov/nchs/nhanes/.

**Funding:** This work was supported by The R&D funds of the second "Jiangbei Talents" mid-term project (No.1391).

**Competing interests:** The authors have declared that no competing interests exist.

**Abbreviations:** ABSI, a body shape index; BMI, body mass index; CIs, confidence intervals; GAM, generalized additive model; HDL-C, high-density lipoprotein cholesterol; IL-6, interleukin-6; IDF, International Diabetes Federation; NHANES, National Health and Nutrition Examination Survey; ORs, odds ratios; SD, standard deviations; TNF-α, tumor necrosis factor-alpha; WC, waist circumference; WHtR, waist-to-height ratio.

## Conclusion

Our findings suggest that there is a positive correlation between diabetes and increased ABSI. ABSI may serve as an effective alternative indicator to other obesity indices, such as BMI.

## Introduction

Diabetes, a global health challenge, is characterized by chronic hyperglycemia resulting from impaired insulin secretion or function [1]. Alarmingly, the World Health Organization estimates that nearly 422 million adults worldwide are afflicted with this condition, with projections anticipating a staggering increase to 700 million by 2045 [2]. The persistent state of uncontrolled hyperglycemia in diabetic patients often gives rise to an array of devastating complications, including cardiovascular diseases, neuropathies, nephropathies, and retinopathies, thereby posing a grave threat to both the quality of life and lifespan [3–6]. Furthermore, diabetes patients often face losses related to reduced productivity, premature death, disability, and inability to work [7–9]. Additionally, diabetes poses not only a severe threat to individual health but also places a substantial burden on societal systems [9,10]. According to the International Diabetes Federation (IDF), global healthcare expenditures related to diabetes reached USD 966 billion in 2021, with projections estimating a rise to USD 1.03 trillion by 2045 [9,10].

In the United States, obesity remains a major public health concern, with more than 40% of adults classified as obese [11,12]. Marked disparities exist across population subgroups, with prevalence approaching 50% among non-Hispanic Black adults and 45% among Hispanic adults, compared with lower rates in non-Hispanic White and Asian populations [11,12]. These patterns underscore both the high burden of obesity in the U.S. and the representativeness of the NHANES cohort for evaluating obesity-related diabetes risk [11,12]. Given the high prevalence of obesity and its strong link to diabetes, accurate assessment of body fat and its distribution is essential [13]. While traditional weight indices such as body mass index (BMI) remain widely used, they often fall short in accurately capturing the distribution of body fat, particularly central adiposity, which is a critical risk factor for metabolic disorders like diabetes [14–16]. To address this limitation, researchers have begun exploring alternative measures that offer a more comprehensive assessment.

One such measure is the A Body Shape Index (ABSI), introduced in 2012 by Krakauer et al. The ABSI combines waist circumference (WC), BMI, and height measurements to provide a risk assessment tool for body shape that is independent of BMI [16]. The ABSI's strength lies in its consideration of the standardized relationship between WC and height and weight, which effectively reflects the degree of abdominal obesity [16]. This has been demonstrated in numerous epidemiological studies, which have shown that the ABSI is independently associated with traditional cardiovascular risk factors such as hypertension and dyslipidemia [17–20]. Moreover,

the ABSI has been found to be a more effective predictor of cardiovascular disease risk compared to both BMI and the waist-to-height ratio (WHtR) [17–20].

Despite these promising findings, the association between the ABSI and diabetes remains understudied. Given the substantial impact of diabetes on both individual and societal health, a more comprehensive understanding of this relationship is imperative. Therefore, the aim of this study is to investigate the relationship between the ABSI and the risk of diabetes using data from the National Health and Nutrition Examination Survey (NHANES) database spanning 1999–2020. By utilizing multivariate adjustments and long-term follow-up data, we hope to provide a more precise and reliable assessment of this relationship. The findings of this study are expected to fill existing gaps in the literature and support the application of the ABSI in the prevention and management of diabetes.

## Materials and methods

### Study participants and research design

Utilizing a stratified multistage random sampling design, NHANES serves as a representative national cross-sectional survey, specifically engineered to mirror the health-related characteristics of the non-institutionalized civilian population in the United States [21]. A detailed description of the NHANES study and its data is available online at https://www.cdc.gov/nchs/nhanes/.

The data for this study were drawn from the NHANES surveys conducted from 1999 to 2020. A total of 107,621 participants were initially selected, of whom 77,417 had relevant data on diabetes. We then excluded 15,441 participants who lacked data on ABSI. Finally, 27,283 individuals under the age of 20 were excluded from the analysis. Consequently, 34,693 participants were included in the study (Fig 1). Participants in the NHANES study signed informed consent forms, and the protocols were approved by the National Center for Health Statistics Research Ethics Review Board. This study used publicly available data and did not require ethics approval.

### Anthropometric measurements

Utilizing standardized techniques and equipment, experienced examiners collected fundamental anthropometric data at mobile examination centers. Measurements included height, BMI, and WC. Notably, WC was measured at the upper margin of the iliac crest [21]. The ABSI was calculated using a formula previously published [16]:

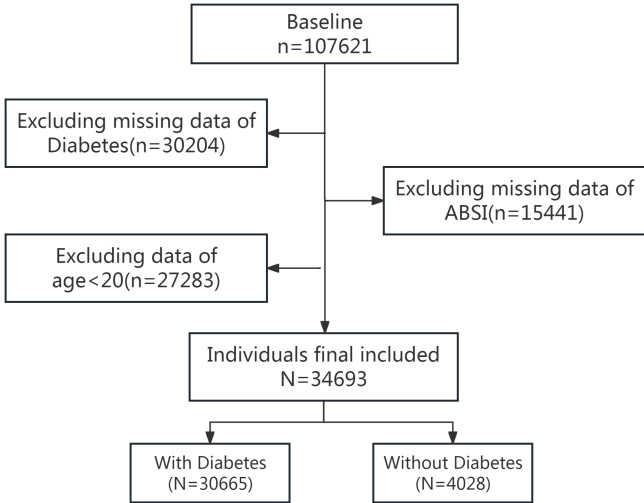

**Fig 1. Flow chart of the screening process for the selection of eligible participants in NHANES 1999–2020.**

$$ABSI = WC/(Height^{1/2} * BMI^{2/3})$$

## Diabetes measurement

Diabetes was the dependent variable in this study. The diagnostic criteria were based on the 2018 American Diabetes Association Standards of Medical Care in Diabetes [22]. Participants meeting one or more of the following criteria were considered to have diabetes: fasting plasma glucose ≥7.0 mmol/L, glycated hemoglobin (HbA1c) ≥6.5%, self-reported diagnosis of diabetes, or current use of glucose-lowering medication [22].

## Covariates

Demographic characteristics and medical history were obtained through a structured questionnaire. Age was used as a continuous variable; gender was categorized into male and female; race was divided into Mexican American, non-Hispanic White, non-Hispanic Black, other Hispanic, and other; marital status was classified into married/living with partner, widowed/divorced/separated, and never married; educational level was divided into less than high school, high school, and more than high school. Participants were categorized as yes or no based on whether they consumed more than 12 alcoholic drinks in the past year. Hypertension, arthritis, congestive heart failure, coronary heart disease status, angina, heart disease, stroke status was classified as yes or no; and levels of high-density lipoprotein cholesterol (HDL-C) were recorded. All details regarding these variables are available on the website at www.cdc.gov/nchs/nhanes/.

## Statistical analysis

The baseline characteristics of the study population were delineated using ABSI and its quartiles. Continuous variables were expressed as means ± standard deviations (SD), while categorical variables were presented as counts and percentages (n (%)). The t-test and chi-square test were employed to evaluate the demographic characteristics of the participants. Odds ratios (ORs) and 95% confidence intervals (CIs) for the association between ABSI and diabetes were estimated through multivariate linear regression analysis. Three models were constructed for the multivariate tests: Model 1: unadjusted; Model 2: adjusted for gender, age, race, educational level, and marital status; Model 3: adjusted for all covariates. Missing data were handled as follows: variables with more than 20% missing values were excluded from the analysis. For categorical variables, missing values were imputed using the mode (i.e., the most frequent category). For continuous variables, if normally distributed, the mean was used for imputation; otherwise, the median was applied. Analysis of non-linear relationships was conducted using a Generalized Additive Model (GAM) with smooth curve fitting and threshold effect analysis.

To examine whether a threshold effect exists in the association between ABSI and diabetes, we used a two-piece linear regression model with a smoothing function. A threshold, or inflection point, was defined as the value of ABSI at which the relationship with diabetes risk changes significantly—statistically identified as the point where segmented regression yields the best model fit [23,24]. The threshold was determined using a two-step recursive approach [23,24]. In the first step, we screened candidate thresholds across the 5th to 95th percentiles of ABSI (at 5% intervals) and selected the percentile that maximized the log-likelihood of the model. In the second step, we narrowed the range around the initial estimate and applied an iterative procedure to identify the precise ABSI value that yielded the highest model likelihood. To compare the goodness of fit between the segmented (two-piece) and standard linear models, we performed a log-likelihood ratio test. A statistically significant result from this test was used as the criterion to support the presence of a threshold effect.

Analysis of non-linear relationships was also applied to gender subgroups. Finally, the stability of the primary outcomes was explored through multifactorial stratified subgroup analysis, with stratification factors including gender (male/

female), race (Non-Hispanic White/Non-Hispanic Black/Mexican American/Other Hispanic/Other), hypertension (yes/no), and stroke (yes/no). To formally assess effect modification, interaction terms between ABSI and each stratification factor (e.g., ABSI × gender) were included in the multivariable logistic regression models. P-values for interaction were calculated using Wald tests for the corresponding interaction terms. These analyses were exploratory in nature and were not pre-specified in the original study design. Statistical analyses were performed using EmpowerStats (http://www.empower-stats.com) and R (version 4.2.2). A two-sided P < 0.05 was considered statistically significant.

## Results

### Baseline characteristics

In this study, adhering to predefined inclusion and exclusion criteria, a total of 34,693 adults were enrolled, with an average age of 50.26 ± 18.32 years. The cohort comprised 50.66% males and 49.34% females. The ABSI quartiles were defined as Tertile 1 (<0.082), Tertile 2 (0.082–0.085), Tertile 3 (0.085–0.089), and Tertile 4 (>0.089). Significant differences were observed across the quartiles in terms of gender, race, educational level, marital status, drinking status, arthritis, congestive heart failure, coronary heart disease, angina, heart disease, and stroke (all P < 0.001). Additionally, compared to the lowest ABSI group, participants in the highest ABSI group were older, had lower income levels and HDL-C levels (all P < 0.001) (Table 1).

### The association between a body shape index and diabetes

Our study results indicate that higher individual ABSI is associated with an increased risk of diabetes. In stepwise adjusted multivariate regression analyses treating ABSI as a continuous variable, this association was significant in both Model 1 (OR = 1.66, 95% CI: 1.59, 1.74) and Model 2 (OR = 1.64, 95% CI: 1.54, 1.73). After full covariate adjustment in Model 3, the positive correlation between ABSI and diabetes remained stable (OR = 1.42, 95% CI: 1.33, 1.52), indicating that each unit increase in ABSI score is associated with a 42% increase in the likelihood of diabetes. Additionally, when ABSI was categorized into quartiles, participants in the highest quartile of ABSI had 96% higher odds of having diabetes compared to those in the lowest quartile (OR = 1.96, 95% CI: 1.69, 2.26) (Table 2), and the upward trend was significant (All p for trend < 0.05).

Additionally, this study employed smooth curve fitting methods and threshold effect analysis to detect potential non-linear relationships between ABSI and diabetes. The results revealed a clear inverse L-shaped curve relationship between ABSI and diabetes (Fig 2). According to the threshold effect analysis, the inflection point for ABSI was determined to be 9.54. The two-part linear regression model demonstrated that when ABSI is less than or equal to 9.54, each unit increase in ABSI is associated with an 87% increased likelihood of diabetes (OR = 1.87, 95% CI: 1.72, 2.05, P < 0.001) (Table 2); however, the relationship was not significant when ABSI was greater than 9.54. When modeling the relationship between ABSI and diabetes, the segmented logistic regression model was superior to the linear logistic regression model (P < 0.001) (Table 3).

### Non-linear positive association of a body shape index and diabetes in males

This study further employed smooth curve fitting to address non-linear issues in gender stratification, revealing a non-linear relationship among male participants (Fig 3). Using the two-part linear regression model, the breakpoint (K) for males was calculated to be 9.43. To the left of the breakpoint, a positive correlation between ABSI and diabetes was observed (OR = 1.84, 95% CI: 1.63, 2.08). However, no statistically significant relationship was observed to the right of the breakpoint (OR = 0.99, 95% CI: 0.78, 1.25) (Table 4).

### Subgroup analysis

By integrating Model 3 for subgroup analyses, the stability of the association between ABSI and the likelihood of having diabetes was further confirmed across different populations (Fig 4). This study conducted subgroup and interaction

**Table 1. Baseline characteristics of the study population according to a body shape index.**

| ABSI[1] | Overall | Tertile 1 | Tertile 2 | Tertile 3 | Tertile 4 | p-value |
|---|---|---|---|---|---|---|
| Age, mean ± sd | 50.26 ± 18.32 | 45.95 ± 18.42 | 50.20 ± 18.20 | 52.00 ± 18.16 | 52.90 ± 17.70 | **<0.001** |
| PIR, mean ± sd | 2.57 ± 1.56 | 2.69 ± 1.61 | 2.73 ± 1.58 | 2.52 ± 1.53 | 2.34 ± 1.49 | **<0.001** |
| **Gender, n (%)** | | | | | | |
| Male | 17577 (50.66%) | 5689 (65.59%) | 5242 (60.44%) | 4017 (46.32%) | 2629 (30.31%) | **<0.001** |
| Female | 17116 (49.34%) | 2984 (34.41%) | 3431 (39.56%) | 4656 (53.68%) | 6045 (69.69%) | |
| **Race, n (%)** | | | | | | |
| Mexican American | 6247 (18.01%) | 729 (8.46%) | 1380 (15.91%) | 1864 (21.49%) | 2274 (26.22%) | **<0.001** |
| Other Hispanic | 2776 (8.00%) | 392 (4.52%) | 635 (7.32%) | 816 (9.41%) | 933 (10.76%) | |
| Non-Hispanic White | 16109 (46.43%) | 4768 (54.98%) | 4334 (49.97%) | 3711 (42.79%) | 3296 (37.99%) | |
| Non-Hispanic Black | 6512 (18.77%) | 1839 (21.20%) | 1486 (17.13%) | 1560 (17.99%) | 1627 (18.76%) | |
| Other Races | 3049 (8.79%) | 945 (10.90%) | 838 (9.66%) | 722 (8.32%) | 544 (6.27%) | |
| **Education level, n (%)** | | | | | | |
| Less than high school | 9704 (27.97%) | 1828 (21.08%) | 2165 (24.96%) | 2617 (30.17%) | 3094 (35.67%) | **<0.001** |
| High school or GED | 7912 (22.81%) | 1926 (22.21%) | 1965 (22.66%) | 2047 (23.60%) | 1974 (22.76%) | |
| Above high school | 17022 (49.07%) | 4908 (53.59) | 4524 (52.16) | 3997 (44.09) | 3593 (41.43) | |
| Unknown | 55 (0.16%) | 12 (0.14%) | 19 (0.22%) | 12 (0.14%) | 12 (0.14%) | |
| **Marital status, n (%)** | | | | | | |
| Married/Living with partner | 20293 (58.49%) | 4823 (55.01%) | 5385 (61.23%) | 5179 (60.06%) | 4906 (57.65%) | **<0.001** |
| Widowed/Divorced/Separated | 7736 (22.30%) | 1648 (18.80%) | 1672 (19.01%) | 2070 (24.01%) | 2346 (27.57%) | |
| Never married | 5812 (16.75%) | 2117 (24.14%) | 1422 (16.17%) | 1149 (13.32%) | 1124 (13.21%) | |
| Missing | 852 (2.46%) | 180 (2.05%) | 314 (3.58%) | 224 (2.61%) | 134 (1.57%) | |
| **Had at least 12 alcohol drinks/1 year, n (%)** | | | | | | |
| No | 9603 (27.68%) | 1784 (20.45%) | 2008 (23.24%) | 2549 (29.36%) | 3262 (37.73%) | **<0.001** |
| Yes | 25069 (72.26%) | 6937 (79.51%) | 6629 (76.72%) | 6126 (70.57%) | 5377 (62.19%) | |
| Missing | 21 (0.06%) | 4 (0.05%) | 4 (0.05%) | 6 (0.08%) | 7 (0.08%) | |
| **High Blood Pressure, n (%)** | | | | | | |
| No | 23694 (68.30%) | 6891 (79.49%) | 6105 (70.45%) | 5684 (65.49%) | 5014 (57.77%) | **<0.001** |
| Yes | 10839 (31.24%) | 1735 (20.01%) | 2530 (29.19%) | 2952 (34.01%) | 3622 (41.73%) | |
| Missing | 160 (0.46%) | 43 (0.5%) | 31 (0.36%) | 43 (0.5%) | 43 (0.5%) | |
| **Arthritis, n (%)** | | | | | | |
| No | 25980 (74.89%) | 7079 (81.65%) | 6715 (77.43%) | 6337 (73.05%) | 5849 (67.41%) | **<0.001** |
| Yes | 8544 (24.63%) | 1552 (17.90%) | 1918 (22.12%) | 2294 (26.45%) | 2780 (32.04%) | |
| Missing | 169 (0.48%) | 39 (0.45%) | 39 (0.45%) | 43 (0.50%) | 48 (0.55%) | |
| **Congestive Heart Failure, n (%)** | | | | | | |
| No | 33584 (96.81%) | 8469 (97.66%) | 8412 (96.99%) | 8386 (96.69%) | 8317 (95.88%) | **<0.001** |
| Yes | 996 (2.87%) | 184 (2.12%) | 229 (2.64%) | 254 (2.93%) | 329 (3.79%) | |
| Missing | 113 (0.32%) | 19 (0.22%) | 32 (0.37%) | 34 (0.38%) | 28 (0.32%) | |
| **Coronary Heart Disease, n (%)** | | | | | | |
| No | 33089 (95.38%) | 8373 (96.55%) | 8243 (95.04%) | 8235 (94.95%) | 8238 (94.97%) | **<0.001** |
| Yes | 1442 (4.16%) | 269 (3.10%) | 387 (4.46%) | 400 (4.61%) | 386 (4.45%) | |
| Missing | 162(0.46) | 30 (0.35%) | 43 (0.50%) | 38 (0.44%) | 51 (0.58%) | |
| **Angina, n (%)** | | | | | | |
| No | 33601 (96.86%) | 8493 (97.94%) | 8409 (96.96%) | 8383 (96.66%) | 8316 (95.87%) | **<0.001** |
| Yes | 967 (2.79%) | 156 (1.80%) | 234 (2.70%) | 252 (2.91%) | 325 (3.75%) | |
| Missing | 125 (0.36%) | 23 (0.27%) | 30 (0.35%) | 39 (0.44%) | 33 (0.38%) | |

*(Continued)*

**Table 1.** (Continued)

| ABSI[1] | Overall | Tertile 1 | Tertile 2 | Tertile 3 | Tertile 4 | p-value |
|---|---|---|---|---|---|---|
| **Heart Attack, n (%)** | | | | | | |
| No | 33147 (95.55%) | 8381 (96.64%) | 8283 (95.50%) | 8259 (95.23%) | 8224 (94.81%) | **<0.001** |
| Yes | 1484 (4.28%) | 282 (3.24%) | 378 (4.33%) | 404 (4.64%) | 420 (4.90%) | |
| Missing | 62 (0.18%) | 10 (0.12%) | 15 (0.17%) | 11 (0.13%) | 26 (0.29%) | |
| **Stroke, n (%)** | | | | | | |
| No | 33407 (96.30%) | 8430 (97.21%) | 8381 (96.63%) | 8330 (96.05%) | 8266 (95.30%) | **<0.001** |
| Yes | 1235 (3.56%) | 231 (2.66%) | 283 (3.26%) | 328 (3.78%) | 393 (4.53%) | |
| Missing | 51 (0.14%) | 12 (0.13%) | 9 (0.10%) | 15 (0.17%) | 15 (0.17%) | |
| HDL-C[2], mmol/L, mean±sd | 1.40±0.42 | 1.51±0.45 | 1.40±0.42 | 1.37±0.41 | 1.34±0.38 | **<0.001** |

[1]ABSI, A body shape index; [2]HDL-C, high density lipoprotein-cholesterol.

**Table 2. Association between a body shape index and diabetes.**

| Variables | Model 1[1] | | | Model 2[2] | | | Model 3[3] | | |
|---|---|---|---|---|---|---|---|---|---|
| **ABSI** | β | 95% CI | P value | β | 95% CI | P value | β | 95% CI | P value |
| **Continuous** | 1.66 | 1.59, 1.74 | **<0.001** | 1.64 | 1.54, 1.73 | **<0.001** | 1.42 | 1.33, 1.52 | **<0.001** |
| **Categories** | OR | 95% CI | P value | OR | 95% CI | P value | OR | 95% CI | P value |
| Tertile 1 | 1.0 | | | 1.0 | | | 1.0 | | |
| Tertile 2 | 1.75 | 1.57, 1.96 | **<0.001** | 1.45 | 1.28, 1.64 | **<0.001** | 1.22 | 1.05, 1.41 | **0.007** |
| Tertile 3 | 2.30 | 2.07, 2.56 | **<0.001** | 1.842 | 1.63, 2.08 | **<0.001** | 1.42 | 1.23, 1.63 | **<0.001** |
| Tertile 4 | 3.18 | 2.87, 3.53 | **<0.001** | 2.71 | 2.40, 3.06 | **<0.001** | 1.96 | 1.69, 2.26 | **<0.001** |
| P for trend | **0.004** | | | **0.019** | | | **0.032** | | |

[1]Non-adjusted model I adjust for: None.

[2]Adjust II model adjust for: sex, age, family poverty income ratio, race, education.

[3]Adjust III model adjust for: sex, age, family poverty income ratio, race, education, hypertension, Marital status, Alcohol, Arthritis, Congestive Heart Failure, Coronary Heart Disease, Angina, Heart Attack, Stroke, and HDL cholesterol.

analyses on the relationship between ABSI and the risk of diabetes based on gender, race, hypertension, and stroke. The results of the subgroup analyses indicated a significant interaction between ABSI and stroke status among participants with diabetes (P for interaction<0.05). In the interaction analyses for gender, race, and hypertension, no significant effects were observed on the association between ABSI and the risk of diabetes across different strata (P for interaction>0.05).

## Discussion

This study focuses on the relationship between ABSI and diabetes, utilizing data from the NHANES database to perform multivariate linear regression analysis, aiming to deepen our understanding of ABSI as a tool for predicting disease risk. As the global prevalence of diabetes continues to rise, developing effective risk prediction and management strategies becomes particularly crucial. Traditional risk assessment indicators like BMI, though widely used, have increasingly shown limitations in predicting diabetes and its complications, especially as they fail to accurately reflect the risk of abdominal obesity [25–28]. The introduction of ABSI offers a new perspective in addressing this issue. Unlike BMI, ABSI considers the ratio of WC to height and weight, more effectively capturing the risk of central obesity, which is vital for the prevention and management of diabetes [16,26].

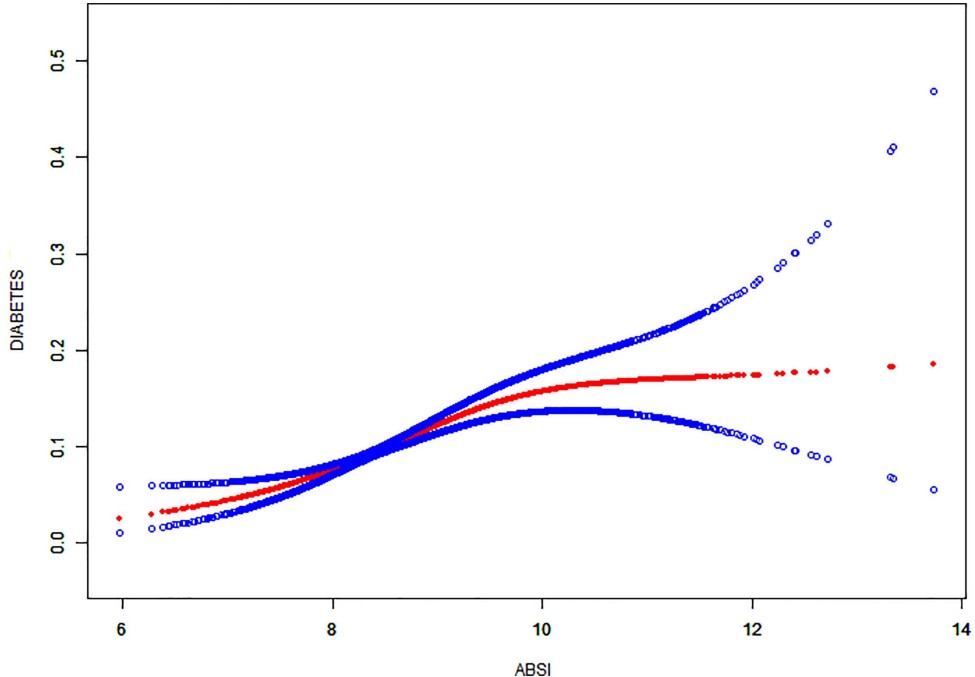

**Fig 2. Adjusted association between ABSI and the predicted probability of diabetes.** The red and blue curves represent multivariable logistic regression estimates with 95% confidence intervals, derived from unweighted data. Models were adjusted for age, gender, race, education level, smoking status, alcohol intake, hypertension, and stroke. X-axis: ABSI (A Body Shape Index); Y-axis: predicted probability of diabetes. Sample size: n = 34,693.

**Table 3. Threshold analysis for the relationship between ABSI and diabetes.**

| Models | Adjusted OR[1] | 95%CI[2] | P value |
|---|---|---|---|
| **Model I** | | | |
| One line slope | 1.50 | 1.41, 1.59 | **<0.001** |
| **Model II** | | | |
| Turning point (K) | 9.54 | | |
| < 9.54 | 1.873 | 1.72, 2.05 | **<0.001** |
| > 9.54 | 0.974 | 0.84, 1.13 | **0.723** |
| OR between < 9.54 and > 9.54 | 0.520 | 0.43, 0.63 | **<0.001** |
| Logarithmic likelihood ratio test | | | **<0.001** |

Adjust for: sex, age, family poverty income ratio, race, education, hypertension, Marital status, Alcohol, Arthritis, Congestive Heart Failure, Coronary Heart Disease, Angina, Heart Attack, Stroke, and HDL cholesterol.

[1]OR: Odds ratio.

[2]95% CI: 95% confidence interval.

In this study, we observed a significant positive correlation between ABSI and diabetes, as demonstrated in Fig 2 and Table 2. Specifically, our results indicate that each unit increase in ABSI is associated with 42% to 66% higher odds of having diabetes, a finding consistently supported across various models adjusted for multiple variables. Moreover, when ABSI was categorized into quartiles, the highest quartile group were nearly 96% more likely to have diabetes compared to the lowest quartile group. These results underscore the potential of ABSI as an independent marker associated with

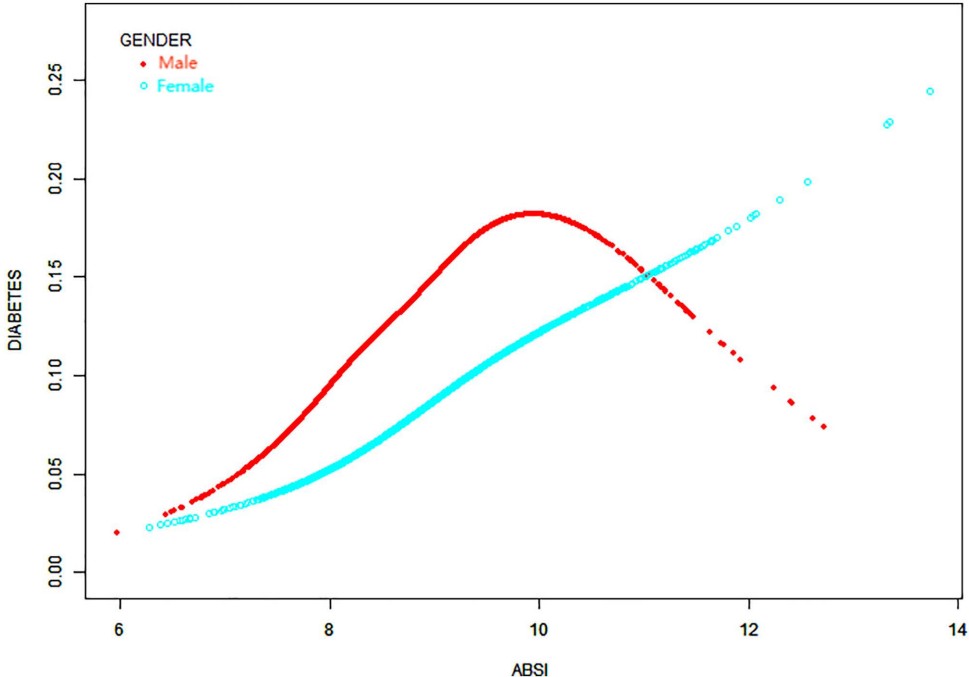

**Fig 3. Adjusted association between ABSI and the probability of diabetes, stratified by gender.** The red and blue smooth curves represent the estimated diabetes probabilities for males and females, respectively, based on multivariable models adjusted for sex, age, family poverty income ratio, race, education, hypertension, Marital status, Alcohol, Arthritis, Congestive Heart Failure, Coronary Heart Disease, Angina, Heart Attack, Stroke, and HDL cholesterol.. The Y-axis indicates the predicted probability of diabetes; the X-axis represents ABSI (A Body Shape Index). Sample sizes: male group (n = 17,577), female group (n = 17,116).

**Table 4. Threshold effect analysis of ABSI on diabetes using a two-piecewise linear regression model.**

| Model I[1] | Male | | | Female | | |
|---|---|---|---|---|---|---|
| | Adjusted OR[3] | 95%CI[4] | P value | Adjusted OR | 95%CI | P value |
| One line slope | 1.53 | 1.41, 1.67 | **<0.001** | 1.470 | 1.36, 1.59 | **<0.001** |
| **Model II[2]** | **Adjusted OR** | **95%CI** | **P value** | **Adjusted OR** | **95%CI** | **P value** |
| Turning point (K) | 9.43 | | | 9.29 | | |
| < K | 1.84 | 1.63, 2.08 | **<0.001** | 2.07 | 1.77, 2.42 | **<0.001** |
| > K | 0.99 | 0.78, 1.25 | 0.90 | 1.08 | 0.93, 1.25 | 0.32 |
| OR between < 9.54 and > 9.54 | 0.54 | 0.39, 0.73 | **<0.001** | 0.52 | 0.40, 0.67 | **<0.001** |
| Logarithmic likelihood ratio test | **<0.001** | | | **<0.001** | | |

[1]Model I Adjust for None.

[2]Model II Adjust for: sex, age, family poverty income ratio, race, education, hypertension, Marital status, Alcohol, Arthritis, Congestive Heart Failure, Coronary Heart Disease, Angina, Heart Attack, Stroke, and HDL cholesterol.

[3]OR: Odds ratio.

[4]95% CI: 95% confidence interval.

diabetes. This finding is corroborated by other similar studies. For example, a study conducted in the Qatari population found that ABSI is a stronger predictor of diabetes risk compared to BMI, aligning with our results and highlighting the potential of ABSI as a disease risk indicator across different ethnicities and regions [29]. It is worth noting that a rural

| Diabetes | | OR | 95%CI | Interaction p−value |
|---|---|---|---|---|
| **Gender** | | | | |
| Male | | 1.404 | (1.240, 1.588) | 0.641 |
| Female | | 1.465 | (1.280, 1.676) | |
| **Ethnicity** | | | | |
| Non−Hispanic White | | 1.54 | (1.245, 1.904) | 0.708 |
| Non−Hispanic Black | | 1.353 | (1.005, 1.821) | |
| Mexican American | | 1.356 | (1.187, 1.548) | |
| Other Hispanic | | 1.496 | (1.243, 1.801) | |
| Other Race | | 1.678 | (1.136, 2.477) | |
| **Stroke** | | | | |
| No | | 1.405 | (1.277, 1.546) | 0.027 |
| Yes | | 2.22 | (1.460, 3.377) | |
| **High Blood Pressure** | | | | |
| No | | 1.357 | (1.202, 1.532) | 0.222 |
| Yes | | 1.52 | (1.331, 1.735) | |

1.0    1.41    2.0    2.83

**Fig 4. Subgroup analysis for the association between ABSI and Diabetes.**

Chinese study also identified a significant association between ABSI and diabetes risk, although ABSI demonstrated relatively lower predictive accuracy compared to other anthropometric indices such as WHtR and BRI, with an AUC of 0.61 in both men and women [30]. In contrast, our study found a stronger association, with individuals in the highest ABSI quartile having a 96% higher odds of diabetes compared to those in the lowest quartile. While we did not perform a direct comparative analysis between ABSI and other indices in our cohort, the findings collectively suggest that ABSI is a relevant marker for diabetes risk in diverse populations. Additionally, two study involving a Japanese adult cohort, which utilized multivariable Cox regression analysis, also identified a correlation between ABSI and the risk of Type 2 diabetes [31,32]. These studies further demonstrated the stability of this association across different population subgroups through subgroup analysis, complementing our findings and indicating the universal applicability of ABSI as a tool for predicting disease risk [31,32]. In summary, despite variations in study design, sample populations, and statistical methods, there is broad support for the potential of ABSI as a predictor of diabetes risk.

ABSI, as a body shape index, is unique in that it combines height, weight, and WC, particularly emphasizing the ratio of WC to other body measurements, thus more accurately reflecting central obesity [16,26]. Central obesity is widely recognized as one of the primary drivers of insulin resistance, a key physiological process in the development of Type 2 diabetes [33]. Specifically, central obesity is associated with the accumulation of abdominal fat, a type of fat characterized by a high rate of fatty acid turnover, which can interfere with insulin signaling pathways through various mechanisms [34–36]. Free fatty acids released by abdominal adipocytes can directly inhibit insulin-mediated glucose uptake and increase hepatic glucose production, thereby reducing systemic insulin sensitivity [36,37]. Additionally, abdominal adipose

tissue serves not merely as a storage site for energy but also functions as an active endocrine organ [38–41]. It is capable of secreting various inflammatory mediators, including tumor necrosis factor-alpha (TNF-α) and interleukin-6 (IL-6) [38–41]. These cytokines can activate inflammatory signaling pathways, further exacerbating insulin resistance [38–41]. Moreover, these factors may also impact the functionality of pancreatic β-cells, accelerating the progression of diabetes [38–41]. Lastly, central obesity is associated with hormonal dysregulation in adipose tissue, such as imbalances in leptin and adiponectin [42–44]. Leptin is typically associated with increased fat storage, and in central obesity, leptin resistance may occur, diminishing its effect on suppressing appetite and potentially disrupting insulin signaling [42]. Simultaneously, a reduction in adiponectin is associated with increased insulin resistance [43,44]. Given that ABSI specifically measures body shape variables related to central obesity, it holds significant potential as a critical predictor of the risk for Type 2 diabetes. It should be noted that the proposed mechanisms involving lifestyle factors and clinical characteristics are based on findings from previous studies and were not directly assessed in our analysis.

This study also identified a non-linear relationship between ABSI and diabetes, manifesting as a distinct inverse L-shaped curve. Through smooth curve fitting and threshold effect analysis, the inflection point of ABSI was determined to be 9.54. Below this inflection point, each unit increase in ABSI was associated with an 87% higher odds of diabetes, whereas above this point, the association was no longer statistically significant. It is important to note that an ABSI value of 9.54, based on the formula incorporating waist circumference, height, and BMI, approximately corresponds to a waist circumference of 106.35 cm in an average adult with a height of 1.70 m and a BMI of 25 kg/m$^2$. This threshold may serve as a practical reference point for identifying individuals at elevated diabetes risk due to central adiposity, even when BMI alone does not appear abnormal. However, this ABSI cut-off and K-breakpoint were derived specifically from the present NHANES cohort, and their applicability to other populations should be interpreted with caution and requires further external validation.

This finding contrasts sharply with another study, which did not detect a threshold effect between ABSI and diabetes [31]. That study observed a consistent positive correlation with increasing ABSI, indicating a linear relationship with Type 2 diabetes risk across the entire observational range, without apparent non-linear characteristics, even after multivariable adjustment [31]. The discrepancy may stem from several factors, particularly differences in the racial and regional composition of the study populations. Our sample, derived from the NHANES database, focused on the American population, whereas the other study involved a Japanese cohort [31]. These racial and geographical differences could account for the significant disparities observed in the relationship between ABSI and diabetes risk [45–50]. Research has shown that Asian populations exhibit distinct characteristics compared to non-Asian populations in terms of fat distribution, insulin sensitivity, and diabetes-related genetic variations, which might affect the sensitivity and specificity of ABSI in predicting diabetes risk [45–47]. Additionally, lifestyle and environmental factors such as dietary habits, physical activity levels, and socioeconomic status also vary across regions, significantly influencing diabetes risk [48–50].

Building on this, the study also employed smooth curve fitting methods to explore non-linear relationships within gender stratification, particularly among male participants. The analysis revealed non-linear characteristics of the relationship between ABSI and the risk of diabetes in the male. Specifically, we identified a breakpoint (K) for male ABSI at 9.43 using a two-part linear regression model. To the left of this breakpoint, when ABSI is below 9.43, there was a significant positive correlation with diabetes risk, showing an 84% increase in diabetes risk for each unit increase in ABSI. However, to the right of the breakpoint, when ABSI exceeded 9.43, no statistically significant relationship was observed between ABSI and diabetes risk. This suggests that at higher levels of ABSI, the predictive power for diabetes risk may diminish or disappear. These findings emphasize the importance of gender and ABSI levels in assessing diabetes. Males demonstrated a higher risk of diabetes at lower ABSI ranges, which may be related to male-specific fat distribution and metabolic characteristics [51–54]. Males tend to accumulate abdominal fat, closely linked to insulin resistance and diabetes risk [51–54]. Additionally, these results indicate that for males, higher ABSI values do not necessarily imply a higher disease risk, which may be due to differences in the distribution and function of adipose tissue at various ABSI levels [51–54]. Overall, these results

provide further evidence supporting the clinical application of ABSI, particularly in male populations, where monitoring at lower ABSI stages may be more critical. This also suggests that the interaction between gender and ABSI levels should be considered when using ABSI for health risk assessments.

Finally, the subgroup analyses conducted in this study further confirmed the stability of the association between ABSI and diabetes. Through analyses of different subgroups, including gender, race, hypertension, and stroke, we found that the relationship between ABSI and the risk of diabetes was consistent across multiple dimensions. Notably, our results identified a significant interaction between ABSI and stroke status, indicating that the association between ABSI and diabetes risk differs between individuals with and without a history of stroke. Specifically, among participants with a history of stroke, the relationship was more pronounced. This may be attributed to stroke-related alterations in metabolic function, such as increased insulin resistance, chronic low-grade inflammation, and changes in body composition due to reduced mobility or muscle mass loss [55–57]. These factors may amplify the impact of central adiposity, as reflected by ABSI, on diabetes risk [55–57]. The presence of this interaction underscores the importance of considering comorbid conditions such as stroke when evaluating the clinical utility of ABSI in diabetes risk assessment and disease management. However, in other subgroups such as gender, race, and hypertension status, the effects of ABSI did not show significant interactions. Another study also conducted similar subgroup analyses and likewise demonstrated a generally positive correlation between ABSI and diabetes [31]. This study found that the stability of the relationship between ABSI and diabetes was higher among females, non-smokers, individuals with fatty liver, and participants with higher BMI, although the tests for interaction did not reveal statistical significance [31]. These findings are consistent with our results, suggesting that ABSI may be a useful indicator associated with diabetes risk across different populations.

## Strengths and limitations

This research has several advantages in exploring the relationship between ABSI and diabetes risk, ensuring the reliability and broad applicability of the results: First, this study utilized the NHANES database, which covers a wide population across the United States. The large-scale sample not only enhanced the statistical power of the research findings but also ensured national representativeness, thereby increasing the generalizability and extrapolation of the study findings. Second, in the multivariate regression models, we adjusted for a variety of potential confounding covariates, including age, gender, race, lifestyle habits, and diseases. This rigorous statistical control helped eliminate potential external interferences that could affect the accuracy of the results, making our findings more reliable and precise. Additionally, a significant discovery of this study is the non-linear relationship between ABSI and diabetes risk. This finding not only provides a new dimension to the understanding of ABSI as a predictive tool for diabetes risk but also challenges the traditional assumption of a linear relationship for such indices, offering a new perspective for future research directions. Lastly, by exploring the relationship between ABSI and diabetes risk across different population subgroups, this study further validated the robustness of its findings. Particularly, considering factors such as gender, race, and chronic disease history, the subgroup analysis results indicated that the association between ABSI and diabetes risk was consistent across all subgroups, emphasizing the wide applicability of ABSI as a risk indicator.

This study has several limitations. First, its cross-sectional design restricts our ability to draw causal inferences between ABSI and diabetes. Although cross-sectional studies can identify associations, they do not determine temporal sequence or causality. Therefore, while we observed a significant association between ABSI and diabetes, we cannot conclude that changes in ABSI directly lead to increased diabetes risk. Second, although we adjusted for a range of potential confounding factors—including age, gender, race, and other sociodemographic and clinical variables—residual confounding cannot be entirely excluded. In particular, alcohol consumption was categorized based on the NHANES coding scheme, using a threshold of more than 12 alcoholic drinks per year. We acknowledge that this definition reflects a relatively low level of intake and does not align with international guidelines, which may limit the interpretability of alcohol-related findings. Unmeasured or unaccounted-for variables may also influence the observed relationship. Finally, the study

population was limited to adults in the United States. As a result, the generalizability of our findings to other countries or ethnic groups may be limited, and further studies in more diverse populations are warranted.

## Conclusion

Overall, this study explored the relationship between ABSI and diabetes. The results indicate a significant non-linear relationship between ABSI and diabetes, particularly evident among males. Additionally, subgroup analyses further confirmed the universality and stability of the relationship between ABSI and diabetes across different populations. These findings highlight ABSI as an potential effective tool for assessing diabetes risk, especially within specific ranges of ABSI values. It provides a valuable risk assessment metric for the clinical and public health fields, aiding in the more precise identification of high-risk individuals and the implementation of targeted preventive measures. However, given the cross-sectional nature of the study and the possibility of residual confounding, causality cannot be inferred. Future longitudinal studies are needed to confirm the temporal sequence and explore causal relationships between ABSI and diabetes, as well as to assess the applicability of these findings across diverse ethnic and cultural groups.

## Acknowledgments

The authors wish to thank all the participants of this study for their important contributions and the National Center for Health Statistics of the Centers for Disease Control and Prevention for sharing the National Health and Nutrition Examination Survey (NHANES) data.

## Author contributions

**Conceptualization:** Danping Zhu, Xijian Zhang.

**Formal analysis:** Zhixi Zhu, Fang Fang.

**Funding acquisition:** Fang Fang.

**Investigation:** Yunhua Li.

**Supervision:** Danping Zhu.

**Validation:** Cai Tang, Zhixi Zhu.

**Visualization:** Yunhua Li.

**Writing – original draft:** Kaiqi Chen.

**Writing – review & editing:** Kaiqi Chen, Cai Tang, Shikui Cui, Zhixi Zhu.

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
