## [Decision Letter · Decision Letter 0]

3 Jun 2025

PONE-D-25-02776Non-linear associations of a body shape index with diabetes among adults: a cross-sectional studyPLOS ONE

Dear Dr. fang,

Thank you for submitting your manuscript to PLOS ONE. After careful consideration, we feel that it has merit but does not fully meet PLOS ONE’s publication criteria as it currently stands. Therefore, we invite you to submit a revised version of the manuscript that addresses the points raised during the review process.

We look forward to receiving your revised manuscript.

Kind regards,

Natural Hoi Sing Chu, Ph.D

Academic Editor

PLOS ONE

2. Thank you for stating the following financial disclosure:  [This work was supported by The R&D funds of the second "Jiangbei Talents" mid-term project (No.1391)].

e) Please provide an amended Funding Statement that declares *all* the funding or sources of support received during this specific study (whether external or internal to your organization) as detailed online in our guide for authors at http://journals.plos.org/plosone/s/submit-now. 

f) Please state what role the funders took in the study.  If any authors received a salary from any of your funders, please state which authors and which funder. If the funders had no role, please state: "The funders had no role in study design, data collection and analysis, decision to publish, or preparation of the manuscript."

Please send your amended statements by return email; we will change the online submission form on your behalf.

Additional Editor Comments (if provided):

Reviewers' comments:

Reviewer's Responses to Questions

**Comments to the Author**

1. Is the manuscript technically sound, and do the data support the conclusions?

Reviewer #1: Partly

Reviewer #2: Yes

2. Has the statistical analysis been performed appropriately and rigorously? 

Reviewer #1: Yes

Reviewer #2: Yes

3. Have the authors made all data underlying the findings in their manuscript fully available?

Reviewer #1: Yes

Reviewer #2: Yes

4. Is the manuscript presented in an intelligible fashion and written in standard English?

Reviewer #1: Yes

Reviewer #2: Yes

5. Review Comments to the Author

Reviewer #1: Major Points Needing Revision:

Causal Language

The manuscript often implies causality ("ABSI increases diabetes risk"), which is not appropriate for a cross-sectional study. Please revise these statements throughout to reflect association, not causation.

The conclusion should explicitly state that longitudinal studies are needed to explore causality.

Threshold Definition & Interpretation

The choice of the ABSI threshold (e.g., 9.54) is automatically selected, but the method is not explained in enough detail. Please clarify how the inflection point was determined, and why the model fit was better with a two-piece model (e.g., include AIC/BIC values or R² comparison).

Explain what the ABSI value of 9.54 represents in practical or clinical terms—e.g., how does this relate to real-world body measurements?

Subgroup Analyses

While subgroup analyses (stroke, gender, race) are reported, interaction terms are mentioned only briefly. Please elaborate on:

How interaction p-values were calculated.

Whether these were pre-specified or exploratory.

The biological plausibility behind the significant interaction with stroke.

Figure and Table Clarity

Figures 2 and 3 need clearer legends, especially explaining what the lines and ribbons represent. Adding sample sizes, axis labels, and whether curves are adjusted or crude is important.

Table 2 and Table 4 are difficult to read due to formatting. Consider splitting or simplifying tables and ensuring that all model covariates are defined in footnotes.

Reviewer #2: Comments to the authors

The authors explore the relationship between the A Body Shape Index (ABSI) and diabetes risk, utilising data from NHANES, and provide valuable insights into the predictive power of ABSI for type 2 diabetes, particularly in the context of central obesity. The study utilises NHANES data to obtain a large, representative sample, ensuring high statistical power and generalisability. It employs multivariate regression models, adjusting for variables including age, gender, race, and pathological factors. A key finding is the non-linear relationship between ABSI and diabetes risk, which challenges linear assumptions and offers new perspectives for risk assessment. Subgroup analyses confirm ABSI’s consistent association with diabetes across gender, race, and chronic disease groups (stroke, heart disease, and heart failure), supporting its broad applicability.

Major comments

1. Page 10 – Could you please clarify the rationale behind recommending at least 12 alcoholic drinks per year, as this represents a low threshold (equivalent to approximately one drink per month) and does not align with standard alcohol consumption categories? There are international guidelines on alcohol consumption that categories alcohol intake, which could be considered to analyse further the impact on ABSI concerning various levels of alcohol intake, if data is available from the dataset.

2. Table 1 - The total number of various baseline characteristics, including education level, marital status, alcohol consumption, high blood pressure, arthritis, coronary heart disease, congestive heart failure, angina, heart attack, and stroke, does not equate to the cohort size (34,693 participants). Please verify the calculations for accuracy and clarify any discrepancies arising from missing data, subgroup exclusions, or reporting errors, ideally by including a footnote or explanation in the methods section.

3. In lines 166-175, please include the reference to Table 2 when the statistics for Model 1 and Model 2 are first mentioned.

4. Lines 248-251 report a percentage increase in ABSI correlating with diabetes risk. Please specify which table or figure supports this finding to ensure clarity and traceability of the reported correlation.

5. Was there further analysis to explore the correlation between ABSI and lifestyle factors (e.g. sleep patterns, stress, and exercise frequency), as well as clinical characteristics (e.g. insulin resistance, beta-cell dysfunction, and other hormonal markers)? Since these factors were highly emphasized in the discussion, further investigation is necessary.

6. The authors reference a rural Chinese study showing similar predictive effectiveness of ABSI for diabetes risk. Could the authors clarify how their findings compare, notably whether ABSI demonstrates superior predictive accuracy (e.g., higher AUC or sensitivity) over other indices like BMI in this or the Chinese cohort?

7. The authors (lines 367–371) underscore the significance of this study by demonstrating that ABSI is a statistically significant predictor of diabetes risk, showing positive correlations with various covariates. However, to establish ABSI as a robust predictive tool, further elaboration will be required on its application, especially for continuous variables. Specifically, the authors should clarify how ABSI’s continuous nature is modelled (e.g. through thresholds or logistic regression) to predict diabetes risk, including details on predictive performance metrics and its added value over existing measures, given the continuous covariates in the analysis.

6. PLOS authors have the option to publish the peer review history of their article (what does this mean? ). If published, this will include your full peer review and any attached files.

**Do you want your identity to be public for this peer review?** For information about this choice, including consent withdrawal, please see our Privacy Policy .

Reviewer #1: **Yes: ** Elabbass Ali Abdelmahmuod

Reviewer #2: No

---

## [Author Response · Author response to Decision Letter 1]

21 Jul 2025

Re: Response for manuscript PONE-D-25-02776 “Non-linear associations of a body shape index with diabetes among adults: a cross-sectional study”

Dear Editor, Dear reviewers

We appreciate your letter dated 3 June , 2025.

We are gratified to have received the peer review comments from Frontiers in Psychiatry. Our sincere thanks go to the editors and reviewers for their invaluable support and guidance in refining our work. We deeply appreciate your constructive feedback and insightful suggestions regarding our paper.

Adhering to the guidelines provided in your letter, we have uploaded both the revised manuscript and a version with tracked changes. All modifications are prominently marked in red font for easy identification.

Enclosed with this letter is our detailed, point-by-point response to the reviewers' comments. We are grateful for the opportunity to resubmit our revised manuscript. It is our sincere hope that Frontiers in Psychology will consider our manuscript suitable for publication.

Sincerely,

The authors

Encl. Responses to the comments from Reviewers 1 and Reviewers 2.

Reply to Reviewers 1

Dear Reviewers,

Thank you for the time and effort you have dedicated to reviewing our manuscript and for your constructive feedback and valuable suggestions.

We greatly appreciate your detailed comments, which have been instrumental in enhancing the quality of our manuscript. We believe that our responses have adequately addressed all of your concerns. In this letter, we will individually address each of your comments and provide our corresponding responses.

We sincerely hope that the revisions made to our manuscript align with your expectations and look forward to your favorable consideration.

Sincerely,

The authors

Major comments:

1. The manuscript often implies causality ("ABSI increases diabetes risk"), which is not appropriate for a cross-sectional study. Please revise these statements throughout to reflect association, not causation.

Author's Response:

We appreciate the reviewer’s insightful comment. In response, we have carefully revised the manuscript to avoid any causal language and to reflect the associative nature of our cross-sectional study. Specifically, phrases such as “increased risk” and “leads to” have been replaced with non-causal alternatives such as “associated with,” “higher odds of,” or “linked to.” These changes have been made throughout the Abstract, Results, and Discussion sections to ensure alignment with the study design.

Modifications:

1)Original sentence: The two-part linear regression model demonstrated that when ABSI is less than or equal to 9.54, each unit increase in ABSI is associated with an 87% increase in the risk of diabetes.

Revised sentence (Line 212-215): The two-part linear regression model demonstrated that when ABSI is less than or equal to 9.54, each unit increase in ABSI is associated with an 87% increased likelihood of diabetes.

2)Original sentence: This study employed multivariate linear regression analysis to evaluate the association between ABSI and the risk of diabetes.

Revised sentence (Line 20-22): This study employed multivariate linear regression analysis to evaluate the association between ABSI and the likelihood of having diabetes.

3)Original sentence: Participants in the highest quartile of ABSI had a 96% greater risk of diabetes.

Revised sentence (Line 28-29): Participants in the highest quartile of ABSI had a 96% higher odds of having diabetes.

4)Original sentence: Each unit increase in ABSI score is associated with a 42% increase in the risk of diabetes.

Revised sentence (Line 195-196): Each unit increase in ABSI score is associated with a 42% increase in the likelihood of diabetes.

5)Original sentence: When ABSI was categorized into quartiles, participants in the highest quartile of ABSI had a 96% increased risk of diabetes compared to those in the lowest quartile.

Revised sentence (Line 197-199): When ABSI was categorized into quartiles, participants in the highest quartile of ABSI had 96% higher odds of having diabetes compared to those in the lowest quartile.

6)Original sentence: Our results indicate that each unit increase in ABSI is associated with a 42% to 66% increase in the risk of diabetes.

Revised sentence (Line 283-285): Our results indicate that each unit increase in ABSI is associated with 42% to 66% higher odds of having diabetes.

7)Original sentence: The highest quartile group exhibited nearly a 96% increased risk of diabetes compared to the lowest quartile group.

Revised sentence (Line 286-288): The highest quartile group were nearly 96% more likely to have diabetes compared to the lowest quartile group.

8)Original sentence: These results underscore the potential of ABSI as an independent predictor of diabetes risk.

Revised sentence (Line 288-289): These results underscore the potential of ABSI as an independent marker associated with diabetes.

2. The conclusion should explicitly state that longitudinal studies are needed to explore causality.

Author's Response:

Thank you for the valuable suggestion. We have revised the conclusion to explicitly state that longitudinal studies are needed to explore the temporal and causal relationship between ABSI and diabetes. The updated conclusion also highlights the limitations of the cross-sectional design and the need for further research in diverse populations.

Modifications:

Line 450-454

However, given the cross-sectional nature of the study and the possibility of residual confounding, causality cannot be inferred. Future longitudinal studies are needed to confirm the temporal sequence and explore causal relationships between ABSI and diabetes, as well as to assess the applicability of these findings across diverse ethnic and cultural groups.

3. Threshold Definition & Interpretation

Author's Response:

Thank you for your valuable comment. In response, we have revised the manuscript to provide a clearer definition and interpretation of the threshold identified in our two-piece linear regression model. We now explicitly define the threshold as the ABSI value at which the association with diabetes risk changes significantly and describe the two-step recursive method used to determine it. Additionally, we clarify the rationale for using a segmented model and the interpretation of the inflection point in both statistical and clinical terms. These revisions have been made to improve clarity and transparency in the methodology and results interpretation.

Modifications:

Line 146-159

To examine whether a threshold effect exists in the association between ABSI and diabetes, we used a two-piece linear regression model with a smoothing function. A threshold, or inflection point, was defined as the value of ABSI at which the relationship with diabetes risk changes significantly—statistically identified as the point where segmented regression yields the best model fit [20, 21]. The threshold was determined using a two-step recursive approach [20, 21]. In the first step, we screened candidate thresholds across the 5th to 95th percentiles of ABSI (at 5% intervals) and selected the percentile that maximized the log-likelihood of the model. In the second step, we narrowed the range around the initial estimate and applied an iterative procedure to identify the precise ABSI value that yielded the highest model likelihood. To compare the goodness of fit between the segmented (two-piece) and standard linear models, we performed a log-likelihood ratio test. A statistically significant result from this test was used as the criterion to support the presence of a threshold effect.

4. The choice of the ABSI threshold (e.g., 9.54) is automatically selected, but the method is not explained in enough detail. Please clarify how the inflection point was determined, and why the model fit was better with a two-piece model (e.g., include AIC/BIC values or R² comparison).

Author's Response:

Thank you for your constructive comment. We have revised the Methods section to provide a more detailed explanation of how the ABSI threshold was determined using a two-step recursive procedure based on log-likelihood optimization. Although AIC, BIC, or R² values were not available, we used a log-likelihood ratio test to compare model fit between the segmented and standard linear models. This methodological clarification has been incorporated without introducing any result-based interpretation in the Methods section.

Modifications:

Line 146-159

To examine whether a threshold effect exists in the association between ABSI and diabetes, we used a two-piece linear regression model with a smoothing function. A threshold, or inflection point, was defined as the value of ABSI at which the relationship with diabetes risk changes significantly—statistically identified as the point where segmented regression yields the best model fit. The threshold was determined using a two-step recursive approach. In the first step, we screened candidate thresholds across the 5th to 95th percentiles of ABSI (at 5% intervals) and selected the percentile that maximized the log-likelihood of the model. In the second step, we narrowed the range around the initial estimate and applied an iterative procedure to identify the precise ABSI value that yielded the highest model likelihood. To compare the goodness of fit between the segmented (two-piece) and standard linear models, we performed a log-likelihood ratio test. A statistically significant result from this test was used as the criterion to support the presence of a threshold effect.

5. Explain what the ABSI value of 9.54 represents in practical or clinical terms—e.g., how does this relate to real-world body measurements?

Author's Response:

Thank you for your valuable comment. In response, we have added an explanation of the clinical relevance of the ABSI threshold of 9.54 in the Discussion section. Specifically, we clarified that this value approximately corresponds to a waist circumference of 106.35 cm in an average adult with a height of 1.70 m and a BMI of 25 kg/m², thereby improving the practical interpretability of our findings.

Modifications:

Line 336-346

This study also identified a non-linear relationship between ABSI and diabetes, manifesting as a distinct inverse L-shaped curve. Through smooth curve fitting and threshold effect analysis, the inflection point of ABSI was determined to be 9.54. Below this inflection point, each unit increase in ABSI was associated with an 87% higher odds of diabetes, whereas above this point, the association was no longer statistically significant. It is important to note that an ABSI value of 9.54, based on the formula incorporating waist circumference, height, and BMI, approximately corresponds to a waist circumference of 106.35 cm in an average adult with a height of 1.70 m and a BMI of 25 kg/m². This threshold may serve as a practical reference point for identifying individuals at elevated diabetes risk due to central adiposity, even when BMI alone does not appear abnormal.

6. While subgroup analyses (stroke, gender, race) are reported, interaction terms are mentioned only briefly. Please elaborate on:How interaction p-values were calculated. Whether these were pre-specified or exploratory.

Author's Response:

Thank you for your thoughtful comment. We have revised the Methods section to provide additional details regarding the interaction analysis. Specifically, we now clarify that interaction terms between ABSI and each stratification factor (e.g., gender, race, stroke, hypertension) were included in multivariable logistic regression models. P-values for interaction were derived using Wald tests. We also note that these analyses were exploratory and not pre-specified in the original study design. These clarifications have been incorporated to improve the transparency of our analytic approach.

Modifications:

Line 160-169

Finally, the stability of the primary outcomes was explored through multifactorial stratified subgroup analysis, with stratification factors including gender (male/female), race (Non-Hispanic White/Non-Hispanic Black/Mexican American/Other Hispanic/Other), hypertension (yes/no), and stroke (yes/no). To formally assess effect modification, interaction terms between ABSI and each stratification factor (e.g., ABSI × gender) were included in the multivariable logistic regression models. P-values for interaction were calculated using Wald tests for the corresponding interaction terms. These analyses were exploratory in nature and were not pre-specified in the original study design.

7. The biological plausibility behind the significant interaction with stroke.

Author's Response:

Thank you for this valuable comment. In response, we have revised the Discussion section to elaborate on the potential biological mechanisms underlying the observed interaction between ABSI and stroke. Specifically, we now discuss how stroke-related metabolic changes—such as increased insulin resistance, chronic inflammation, and altered body composition—may enhance the impact of central adiposity on diabetes risk. These additions aim to strengthen the biological plausibility of our findings and highlight the importance of considering comorbid conditions in ABSI-based risk assessment.

Modifications:

Line 389-339

Notably, our results identified a significant interaction between ABSI and stroke status, indicating that the association between ABSI and diabetes risk differs between individuals with and without a history of stroke. Specifically, among participants with a history of stroke, the relationship was more pronounced. This may be attributed to stroke-related alterations in metabolic function, such as increased insulin resistance, chronic low-grade inflammation, and changes in body composition due to reduced mobility or muscle mass loss. These factors may amplify the impact of central adiposity, as reflected by ABSI, on diabetes risk. The presence of this interaction underscores the importance of considering comorbid conditions such as stroke when evaluating the clinical utility of ABSI in diabetes risk assessment and disease management.

8. Figures 2 and 3 need clearer legends, especially explaining what the lines and ribbons represent. Adding sample sizes, axis labels, and whether curves are adjusted or crude is important.

Author's Response:

Thank you for this helpful comment. We have revised the legends of Figures 2 and 3 to clearly describe the meaning of the lines and shaded areas, indicating whether the curves are based on adjusted or unadjusted models. Axis labels have also been clarified to specify the variables plotted and their interpretation, and sample sizes have been added for relevant subgroups. Additionally, we have converted all figures to high-resolution TIFF format to improve image clarity and ensure publication-quality presentation.

Modifications:

Line 221-225

Fig. 2 Adjusted association between ABSI and the predicted probability of diabetes. The red and blue curves represent multivariable logistic regression estimates with 95% confidence intervals, derived from unweighted data. Models were adjusted for age, gender, race, education level, smoking status, alcohol intake, hypertension, and stroke. X-axis: ABSI (A Body Shape Index); Y-axis: predicted probability of diabetes. Sample size: n = 34,693.

Line 243-249

Fig. 3 Adjusted association between ABSI and the probability of diabetes, stratified by gender. The red and blue smooth curves represent the estimated diabetes probabilities for males and females, respectively, based on multivariable models adjusted for sex, age, family poverty income ratio, race, education, hypertension, Marital status, Alcohol, Arthritis, Congestive Heart Failure, Coronary Heart Disease, Angina, Heart Attack, Stroke, and HDL cholesterol.. The Y-axis indicates the predicted probability of diabetes; the X-axis represents ABSI (A Body Shape Index). Sample sizes: male group (n = 17,577), female group (n = 17,116).

10. Table 2 and Table 4 are difficult to read due to formatting. Consider splitting or simplifying tables and ensuring that all model covariates are defined in footnotes.

Author's Response:

Thank you for your valuable suggestion. We have

---

## [Decision Letter · Decision Letter 1]

13 Aug 2025

PONE-D-25-02776R1Non-linear associations of a body shape index with diabetes among adults: a cross-sectional studyPLOS ONE

Dear Dr. fang,

Thank you for submitting your manuscript to PLOS ONE. After careful consideration, we feel that it has merit but does not fully meet PLOS ONE’s publication criteria as it currently stands. Therefore, we invite you to submit a revised version of the manuscript that addresses the points raised during the review process.

We look forward to receiving your revised manuscript.

Kind regards,

Natural Hoi Sing Chu, Ph.D

Academic Editor

PLOS ONE

**Journal Requirements:**

Reviewers' comments:

Reviewer's Responses to Questions

**Comments to the Author**

1. If the authors have adequately addressed your comments raised in a previous round of review and you feel that this manuscript is now acceptable for publication, you may indicate that here to bypass the “Comments to the Author” section, enter your conflict of interest statement in the “Confidential to Editor” section, and submit your "Accept" recommendation.

Reviewer #2: All comments have been addressed

2. Is the manuscript technically sound, and do the data support the conclusions?

Reviewer #2: Yes

3. Has the statistical analysis been performed appropriately and rigorously? 

Reviewer #2: Yes

4. Have the authors made all data underlying the findings in their manuscript fully available?

Reviewer #2: Yes

5. Is the manuscript presented in an intelligible fashion and written in standard English?

Reviewer #2: Yes

6. Review Comments to the Author

**Reviewer #2: ** The revised manuscript presents a more comprehensive analysis of the association between A Body Shape Index (ABSI) and diabetes using NHANES data. The authors have responded promptly to prior reviewers’ feedback, and most previous concerns have been addressed adequately. However, some minor adjustments are still required to enhance the clarity of the manuscript.

1. The number within the arthritis subgroup should be rechecked for accuracy, as prior concerns remain a mismatch between reported subgroup counts and the overall cohort size (34,693).

2. It is advised to include a brief contextual statement on U.S. obesity prevalence among other population subgroups in the introduction or discussion sections. This would help establish the representativeness of the NHANES cohort for international readers and emphasise the importance of the findings related to obesity-associated diabetes risk in the American adult population.

3. Although the manuscript clearly states that the ABSI value and K-breakpoint are derived from this study, it is recommended to briefly reiterate this in the Discussion sections to prevent readers from assuming the cut-off applies to other populations without validation.

7. PLOS authors have the option to publish the peer review history of their article (what does this mean? ). If published, this will include your full peer review and any attached files.

**Do you want your identity to be public for this peer review?** For information about this choice, including consent withdrawal, please see our Privacy Policy .

Reviewer #2: No

---

## [Author Response · Author response to Decision Letter 2]

15 Sep 2025

Dear Editor, Dear reviewers

We appreciate your letter dated 3 June , 2025.

We are grateful to have received the second-round peer review comments from PLOS ONE. Our heartfelt thanks go to the editors and reviewers for their valuable time, constructive feedback, and insightful suggestions, which have been instrumental in further improving the quality and clarity of our work.

In accordance with the guidelines provided, we have submitted both a clean revised version of the manuscript and a tracked-changes version. All modifications made in response to the reviewers’ comments are clearly highlighted for ease of reference.

Enclosed with this letter is our detailed, point-by-point response to the second-round comments. We greatly appreciate the continued guidance and support, and we respectfully hope that PLOS ONE will now find our revised manuscript suitable for publication.

Sincerely,

The authors

Encl. Responses to the comments from Reviewers 2.

Reply to Reviewers 2

Dear Reviewers,

We would like to express our sincere gratitude for your thoughtful and constructive comments on our manuscript during the second round of review. Your careful evaluation and insightful suggestions have been invaluable in refining and strengthening our work.

We have carefully considered each of your comments and have made the corresponding revisions to the manuscript. A detailed, point-by-point response outlining the changes is enclosed for your review. We believe that these revisions have substantially improved the clarity, accuracy, and overall quality of our study.

We are truly appreciative of the time and effort you have devoted to reviewing our work, and we thank you again for your valuable contributions.

Sincerely,

The authors

Major comments:

1. The number within the arthritis subgroup should be rechecked for accuracy, as prior concerns remain a mismatch between reported subgroup counts and the overall cohort size (34,693).

Author's Response:

We sincerely thank the reviewer for carefully checking the subgroup counts. Upon re-examination, we identified that the discrepancy was due to an error in reporting the number of participants in the Missing category for the arthritis variable. In our original submission, 100 individuals were inadvertently omitted from the Missing row. We have now corrected this error. The updated counts and percentages have been revised accordingly in the table and manuscript.

We appreciate the reviewer’s attention, which helped us ensure the accuracy and consistency of the data presentation.

Modifications:

Table 1

Arthritis,

n (%) Overall Tertile 1 Tertile 2 Tertile 3 Tertile 4

No 25980

(74.89%) 7079 (81.65%) 6715 (77.43%) 6337 (73.05%) 5849 (67.41%)

Yes 8544 (24.63%) 1552 (17.90%) 1918 (22.12%) 2294 (26.45%) 2780 (32.04%)

Missing 169 (0.48%) 39 (0.45%) 39 (0.45%) 43 (0.50%) 48 (0.55%)

2. It is advised to include a brief contextual statement on U.S. obesity prevalence among other population subgroups in the introduction or discussion sections. This would help establish the representativeness of the NHANES cohort for international readers and emphasise the importance of the findings related to obesity-associated diabetes risk in the American adult population.

Author's Response:

We appreciate the reviewer’s valuable suggestion to provide additional context on U.S. obesity prevalence. Specifically, we now highlight that more than 40% of U.S. adults are classified as obese, with the highest prevalence observed among non-Hispanic Black and Hispanic adults. This addition underscores the substantial public health burden of obesity in the United States and helps clarify the representativeness of the NHANES cohort for international readers.

Modifications:

Line 58-66

In the United States, obesity remains a major public health concern, with more than 40% of adults classified as obese [11, 12]. Marked disparities exist across population subgroups, with prevalence approaching 50% among non-Hispanic Black adults and 45% among Hispanic adults, compared with lower rates in non-Hispanic White and Asian populations [11, 12]. These patterns underscore both the high burden of obesity in the U.S. and the representativeness of the NHANES cohort for evaluating obesity-related diabetes risk [11, 12]. Given the high prevalence of obesity and its strong link to diabetes, accurate assessment of body fat and its distribution is essential [13].

3. Although the manuscript clearly states that the ABSI value and K-breakpoint are derived from this study, it is recommended to briefly reiterate this in the Discussion sections to prevent readers from assuming the cut-off applies to other populations without validation.

Author's Response:

We thank the reviewer for this important observation. In response, we have revised the Discussion section to reiterate that the ABSI cut-off (9.54) and K-breakpoint identified in our analysis were derived specifically from the NHANES cohort. We now emphasize that their applicability to other populations should be interpreted with caution and requires further external validation. This clarification is intended to prevent readers from assuming the cut-off can be directly generalized beyond the study population.

Modifications:

Line 355-358

However, this ABSI cut-off and K-breakpoint were derived specifically from the present NHANES cohort, and their applicability to other populations should be interpreted with caution and requires further external validation.

---

## [Editor Report · Decision Letter 2]

22 Sep 2025

Non-linear associations of a body shape index with diabetes among adults: a cross-sectional study

PONE-D-25-02776R2

Dear Dr. fang,

We’re pleased to inform you that your manuscript has been judged scientifically suitable for publication and will be formally accepted for publication once it meets all outstanding technical requirements.

Kind regards,

Natural Hoi Sing Chu, Ph.D

Academic Editor

PLOS ONE
---

## [Editor Report · Acceptance letter]

PONE-D-25-02776R2

PLOS ONE

Dear Dr. Fang,

I'm pleased to inform you that your manuscript has been deemed suitable for publication in PLOS ONE. Congratulations! Your manuscript is now being handed over to our production team.

Kind regards,

on behalf of

Dr. Natural Hoi Sing Chu

Academic Editor

PLOS ONE